# Serum Vitamin D Concentration Is Lower in Patients with Tinnitus: A Meta-Analysis of Observational Studies

**DOI:** 10.3390/diagnostics13061037

**Published:** 2023-03-08

**Authors:** Riccardo Nocini, Brandon M. Henry, Camilla Mattiuzzi, Giuseppe Lippi

**Affiliations:** 1Unit of Otorhinolaryngology, Department of Surgery, Dentistry, Paediatrics and Gynaecology, University of Verona, Piazzale L. A. Scuro, 37134 Verona, Italy; 2Clinical Laboratory, Division of Nephrology and Hypertension, Cincinnati Children’s Hospital Medical Center, 3333 Burnet Ave., Cincinnati, OH 45229, USA; 3Medical Direction, Rovereto Hospital, Provincial Agency for Social and Sanitary Services (APSS) of Trento, Via Alcide Degasperi, 38123 Trento, Italy; 4Section of Clinical Biochemistry, School of Medicine, University of Verona, Piazzale L. A. Scuro, 37134 Verona, Italy

**Keywords:** Vitamin D, 25OH-D, cholecalciferol, ergocalciferol, hydroxycholecalciferol, tinnitus

## Abstract

Background: Tinnitus is a highly prevalent and frequently disabling condition, such that the identification of possible causal mechanisms would yield significant clinical and social benefits. Since vitamin D (Vit D) is involved in the pathogenesis of several ear disturbances, we review here the current scientific literature addressing the relationship between Vit D status and tinnitus. Methods: An electronic search was conducted in PubMed, Scopus and Web of Science with the keywords “tinnitus” and “Vitamin D” or “Vit D” or “25OH-D” or “cholecalciferol” or “ergocalciferol” or “hydroxycholecalciferol”, without date (i.e., up to 8 February 2023) or language restrictions, in accordance with a protocol based on the transparent reporting of systematic reviews and meta-analysis (PRISMA) 2020 checklist, for identifying studies which assayed serum Vit D concentration in patients with or without tinnitus. Results: Three observational, case-control studies encompassing four cohorts and totaling 468 patients with (*n* = 268) or without tinnitus (*n* = 200) were included in this meta-analysis. Pooled analysis with quality effects models evidenced significantly reduced serum Vit D levels in patients with tinnitus compared to those without (weighted mean difference [WMD], −6.2 ng/mL; 95% CI, −10.3 to −2.1 ng/mL; I^2^, 56%). Serum Vit D was found to be 22% lower in patients with tinnitus compared to those without. Conclusions: Lower serum Vit D levels may be associated with tinnitus, thus paving the way to plan future trials aimed at exploring whether Vit D supplementation may aid in preventing and/or improving tinnitus.

## 1. Introduction

Tinnitus, a term deriving from the Latin word “tinnire” (i.e., “to ring”) is conventionally defined as perception of a particular sound (like ringing, buzzing, roaring, clicking, hissing, humming, whooshing, throbbing, etc.) in the lack of vibration of an external elastic body, which can be perceived as subjective or objective (i.e., can be heard by an outside observer), pulsatile (e.g., most often heart rhythmic) or not [1,2]. According to recent data, the different forms of tinnitus have a dramatically high burden in the general population, with annual incidence in adults ranging between 1–14% (2% with severe forms) and prevalence of 10% in young adults, increasing to 14% in middle-aged adults, and peaking at 24% in older adults (around 2.3% with severe phenotype), respectively [3]. The burden of this condition has also consistently increased during the coronavirus disease 2019 (COVID-19) pandemic [4], due to direct viral injury of sensorineural hearing apparatus [5], compounded by a considerable onset of COVID-19-realted psychosocial conditions in the general population (e.g., stress, anxiety and depression) that may have worsened a pre-existing tinnitus [6]. This epidemiologic data portrays the picture of a serious public health issue, since the consequences on the daily quality of life of the people affected by permanent (e.g., long-lasting or even chronic) tinnitus may be devastating, encompassing hyperacusis, concentration and communication derangements, annoyance, irritability, depression, anxiety, sleep disturbances and insomnia [7], up to development of suicidal thoughts needing urgent psychiatric interventions [8].

The pathogenesis of tinnitus is complex and often multifactorial, recognizing pathologies of the outer ear (excessive earwax, tympanic membrane injuries or infections), middle ear (i.e., acute or chronic infections, otosclerosis, injuries due to heavy noise exposure, ototoxic drugs usage, middle ear tumors such as glomus tympanicum, muscle spasms, Eustachian tube dysfunction), inner ear (Meniere disease, cochlear injuries, age-related hearing loss or presbycusis), acoustic nerve pathologies (vestibular schwannoma, acoustic neuroma, conflict with itracranial arteries), as well as a kaleidoscope of other causal factors, some located in the nearby tissues (e.g., disorders or malformations of blood vessels, ostemalacia, Paget’s disease, cerebellopontine-angle tumors, temporomandibular joint disorder), others relatively far from the hearing apparatus such as hyperactivity of auditory brain neurons, multiple sclerosis, idiopathic intracranial hypertension, dural blood vessels abnormalities, head and/or neck injuries, musculoskeletal cervical imbalance, anemia, hypertiroidism and hypertension, along with somatoform or phobic disorders [9,10,11]. Notably, although it is important to remember that tinnitus is always a symptom of an underlying pathology and not a disease in itself, the clinical cause(s) or the triggering factor(s) often remain conjectural or even completely unidentifiable [12].

Reliable epidemiologic evidence has been recently provided that lower serum vitamin D (Vit D) levels may be associated with hearing impairment and/or sensory-neural hearing loss [13,14], and balance disorders [15], thus persuading us to conduct a systematic literature search and meta-analysis to explore whether an epidemiologic association may exist between low serum Vit D status and tinnitus.

## 2. Materials and Methods

### 2.1. PRISMA Guidelines

This systematic literature review and meta-analysis was conducted following a protocol based on the transparent reporting of systematic reviews and meta-analysis (PRISMA) 2020 checklist (Appendix A).

### 2.2. Search Strategy

We conducted an electronic search in Medline (using the PubMed interface), Scopus and Web of Science (WoS), with the keywords “tinnitus” and “Vitamin D” or “Vit D” or “25OH-D” or “cholecalciferol” or “ergocalciferol” or “hydroxycholecalciferol”, without date (i.e., up to 10 November 2022) or language restrictions. Title, abstract and full text of all documents that we could first identify based on the aforementioned search criteria were systematically screened by two authors (R.N. and G.L.), and those reporting the results of studies which investigated serum Vit D levels in patients with or without tinnitus were finally included in our analysis. The reference list of all pertinent articles was also hand-searched by means of forward and backward citation tracking, for retrieving additional and potentially eligible documents. 

### 2.3. Statistical Analysis

We carried out a meta-analysis of pertinent studies for estimating the weighted mean difference (WMD) and its 95% confidence interval (95% CI) of serum Vit D levels in subjects with or without tinnitus. The pooled analysis was conducted using both the quality and the random effects models; this latter approach was used for adjusting for possible heterogeneity, which was calculated with χ^2^ test and I^2^ statistics, whilst the risk of publication bias was assessed with funnel plots. The statistical analysis was performed using MetaXL, software Version 5.3 (EpiGear International Pty Ltd., Sunrise Beach, Australia). This study was conducted in accordance with the declaration of Helsinki and within the terms of local legislations. The investigation was exempted from ethical committee approval as it is not locally required for pooled analyses, nor received any funding.

## 3. Results

### 3.1. Study Identification and Selection

After excluding replicate publications among the three scientific search platforms, a total of 72 articles were originally detected using the predefined criteria and by hand-searching the reference lists, 69 of which ought to be eliminated because they did not present a comparison of serum Vit D levels in patients with or without tinnitus (*n* = 29), did not assessed tinnitus (*n* = 16) or Vit D status (*n* = 5), lacked a control group of subjects without tinnitus (*n* = 3), were review articles (*n* = 11), editorial material (*n* = 3) or case reports (*n* = 2). A final number of 3 studies (all observational, case-control), with four cohorts and totaling 468 patients with (*n* = 268) or without tinnitus (*n* = 200) were finally included in our meta-analysis [16,17,18]. The main characteristics of these studies are summarized in Table 1. One study (with two cohorts) was conducted in Iran, one in the Dominican Republic and one in Poland. The sample size was very heterogeneous, varying between 44 and 300, as was the prevalence of tinnitus (i.e., 11.4–69.4%). In one study the method used for measuring Vit D has been reported (i.e., enzyme-linked immunosorbent assay; ELISA), whilst in the remaining two studies this information was lacking.

### 3.2. Meta-Analysis

In all three studies and four cohorts, the serum value of Vit D was found to be lower in patients with tinnitus than in those without, with WMDs ranging between −1.45 to −7.57 ng/mL. The pooled analysis performed using the quality effects models confirmed a significantly negative WMD of Vit D concentration in patients with tinnitus compared to those without, with a WMD of −6.2 ng/mL (95% CI, −10.3 to −2.1 ng/mL; I^2^, 56%), though such difference was largely determined by the study of Nowaczewska et al. [18], expressing the largest sample size (*n* = 300) (Figure 1). 

Overall, the serum Vit D concentration was hence found to be 22% lower in patients with tinnitus compared to those without. A slightly lower but still significant difference was also found using the random effects model (WMD, −4.6 ng/mL; 95% CI, −8.0 to −1.21 ng/mL). The funnel plot, as shown in Figure 2, did not reveal a substantial publication bias.

## 4. Discussion

There is increasing evidence that certain nutritional deficiencies, thus including lower levels of Vit D, may play an important role in the risk of developing hearing impairment and related consequences, one of which is indeed tinnitus.

One of the first studies that underpinned a potential association between Vit D deficiency and impairment of sensory-neural hearing system was published by Gerald B. Brookes, in 1983 [19]. Briefly, this author described the case of ten patients with bilateral cochlear deafness, who were also found to be Vit D deficient. Two years later, the same author reported other 27 patients affected by bilateral deafness and concomitant Vit D deficiency [20]. Notably, since cochlea demineralization resulting in serious morphological changes and impaired neurosensoral hearing transmission was identified as the underlying cause, Vit D replacement therapy was initiated, yielding to hearing improvement in 50% of patients in whom the treatment response become available. In the same year Gerald B. Brookes also noted that Vit D deficiency was commonplace in patients with otosclerosis, causing impairment of cochlear structure and deafness [21]. Although in none of these articles the association between tinnitus and Vit D status was explored, the evidence that this important vitamin would interplay with hearing fitness had been unraveled. It is hence not surprising that a number of very recent studies have highlighted that Vit D deficiency may have causal associations with a kaleidoscope of pathologies which may then evolve, or be causally associated with, tinnitus. Salamah et al. conducted a systematic literature review and meta-analysis to explore the potential association between serum Vit D level and the risk of developing otitis media [22]. A pooled analysis of eleven studies (totaling over 17,000 patients) revealed that the levels of Vit D were significantly lower in patients with both acute (mean difference: −10.6; 95% CI, −19.3 to −2.0) and chronic (mean difference: −3.6; 95% CI, −7.0 to −0.2) otitis media compared to the healthy control population, yielding to a pooled mean difference of −6.26 (95% CI, −10.5 to −2.0) in all patients with otitis media. Such relationship between low Vit D and otitis media may hence justify a concomitantly increased risk of developing tinnitus, as shown in the meta-analysis published by Biswas et al. [23], who concluded that patients with otitis media have an over 60% enhanced risk of developing tinnitus (relative risk [RR], 1.63; 95% CI, 1.61–1.65). Besides otitis, low Vit D levels may also be causally associated with benign paroxysmal positional vertigo. In a recent meta-analysis, published by Yang et al. and including 18 studies with 1859 cases and 1495 controls [24], the authors concluded that Vit D levels were significantly lower in patients with benign paroxysmal positional vertigo compared to control (mean difference: −2.5; 95% CI, −3.79 to −1.1). Identical evidence emerged from the meta-analysis of Chen et al. [25], including 14 studies and 3060 patients with benign paroxysmal positional vertigo. Specifically, the authors found that those with recurrence of paroxysmal positional vertigo had a significantly lower level of Vit D (mean difference: −3.3; 95% CI, −5.3 to −1.3). In analogy with otitis media, patients with benign paroxysmal positional vertigo also have a substantial risk of developing tinnitus, as shown in the meta-analysis published Jafari et al. in 2022 (event rate: 12.2%; 95% CI, 7.0–20.4%) [26]. Various studies recently reviewed by Taneja demonstrated a significant association between nutritional deficiencies, including low Vit D levels, and old age deafness and/or presbycusis [27]. Importantly, Vit D supplementation has also been linked with encouraging results in ameliorating aging deafness. Accordingly, Nondahl et al. found that each 5 dB increase in pure-tone average was associated with a 17% higher risk of developing tinnitus (OR, 1.17, 95% CI, 1.13–1.22) [28].

Although we could only find a limited number of observational studies linking serum Vit D with presence or absence of tinnitus (*n* = 3, with four cohorts, with one including 300 out of 468 individuals), the results emerged from of our meta-analysis reveal that serum Vit D levels displayed a decreasing trend in all such studies in patients with tinnitus compared to those without (Figure 1). Overall, we estimated that serum Vit D levels could be 22% lower in patients with tinnitus, thus frequently encompassing values still comprised within the definition of “insufficiency” (i.e., between 20–30 ng/mL), rather the falling into the straightforward definition of “frank deficiency” (i.e., <20 ng/mL, or even below12 ng/mL). No relevant publication bias could be estimated by our analysis, thus pointing out a definite role for Vit D status in development, perception and/or amplification of this seriously debilitating hearing disturbance.

One article that was excluded from our analysis because no final data on Vit D status in patients with tinnitus were presented deserves special mention. Briefly, the authors administered a questionnaire to 34,576 UK adults aged between 40–69 years to garner information on their nutritional status. Vit D intake was defined as quintiles of dietary patterns, from low to high. In a final regression models including several demographical variables, use of ototoxic therapy, noise exposure, alcohol consumption and cardiovascular disease, subjects in the highest quintile of Vit D intake did not display a significantly different odds of tinnitus compared to those in the lowest quintile of Vit D intake (OR, 0.99; 95% CI, 0.88–1.11; *p* = 0.535), whilst a higher intake of Vit D was found to be associated with lower odds of hearing difficulties (1st vs. 5th quintiles of Vit D intake: OR, 0.90; 95% CI, 0.81–1.00; *p* = 0.013) [29]. Yet, serum Vit D was not measured in patients with or without tinnitus, such that it cannot be assessed to what extent Vit D status may have impacted tinnitus development or perception in this study.

We could also identify another interesting study, which did not compare vitamin status in patients with or without tinnitus, but still presented interesting findings [30]. Briefly, the author assessed Vit D status in 35 adult subjects with bilateral tinnitus (age range, 20–50 years), who were supplemented with oral Vit D (50,000 IU/week) for 3 months. After completing the supplementation period, the tinnitus handicap inventory (THI; a self-reported, 25-item questionnaire to assess the severity of perceived tinnitus handicap) substantially decreased by nearly 40%, from 2.50 ± 0.88 to 1.47 ± 0.57 (*p* < 0.001).

Based on our findings, we propose that several aspects in Vit D deficiency may actually contribute to enhance the risk of developing or worsening tinnitus, as summarized in Table 2.

One of the most obvious mechanisms linking Vit D deficiency to hearing problems encompassed the development of rickets and/or osteomalacia affecting the osteoskeletal system, including skull bones [31]. Thus, besides cochlear demineralization and the resulting neurosensoral hearing transmission impairment which is per se a major cause of tinnitus [19], Vit D-related demineralization of petrous temporal bone may reduce the perception of external (environmental) sounds, enhancing internal resonance and transmission of internal sounds caused by voice, respiration or vascular pulsation among others, thus ultimately triggering tinnitus [32]. This is especially true if one considers that the woven bone of the optic capsule contains a considerably high concentration of calcium [30], such that an impairment of Vit D metabolism may have a profound and unfavorable impact on adequate mineralization of this skeletal district.

Then, Vit D deficiency is associated with an increased risk of developing a vast array of pathologies of the hearing system such as acute and chronic otitis [22], tympanosclerosis [33], otosclerosis [21], but also predisposes to accelerated deafness and presbycusis [27,34]. A strict relationship has been recently underpinned between stress, anxiety and depression-like behaviors [35,36], in that patients with low Vit D serum levels were found to be at enhanced risk of developing these psychophysiological disorders. In turn, an increased burden of stress, anxiety and depression could act by directly triggering newly onset tinnitus, or even by amplifying a pre-existing hearing disturbance [7]. Notably, the relationship between tinnitus and depression is especially important, since it follows a bi-directional path, where depression may predispose to development or intensification of tinnitus, whilst onset or aggravation of tinnitus may then worsen depression, thus generating a devastating biological and psychological loop.

It should then be considered that Vit D deficiency may be a bystander rather than an active player in the complex pathogenesis of tinnitus. For example, Vit D deficiency is commonplace in patients with extremes of body weight, thus in those with malnutrition [37], as well as in those with overweight or obesity [38]. In turn, tinnitus appears to be more prevalent in overweight/obese patients (e.g., due to pseudotumor cerebri syndrome or other disturbances) [39,40], as well as in those underweight [41] and/or with recent weight loss [42], in whom a reduced fat tissue lining may predispose a major propagation of internal sounds to the cochlea or amplify bone-conduction sounds [43]. Finally, Vit D has been convincingly linked to an enhanced risk of developing hypertension [44], since a recent meta-analysis emphasized that hypertensive patients have an increased odd of tinnitus (OR, 1.37; 95% CI: 1.16–1.62) [45].

The study of Fanimolky et al. deserves a particular mention [17]. The authors studied 62 patients with middle ear cholesteatoma and 62 chronic otitis media, 62 of whom tinnitus, and reported modestly decreased Vit D levels in tinnitus patients belonging to both cohorts (16 ± 8 vs. 17 ± 11 ng/mL and 36.1 ± 9.3 vs. 38.6 ± 13.4 ng/mL, respectively). Although the association of otitis and tinnitus is rather clear and intuitive (as early discussed), that between ear cholesteatoma (i.e., a skin-linked, cyst-like structure developed behind the eardrum and potentially extending to the middle ear and mastoid) is especially intriguing. The underlying mechanism encompasses turbulence in the blood flow nearby the hearing apparatus, which could be caused by various conditions leading to increased venous flow or blood vessel stenosis, including cholesteatoma [46]. The sound generated by such turbulent flow can hence be perceived as (mostly pulsatile) tinnitus by the patients.

The results of our meta-analysis may have some potentially useful clinical implications. First, the evidence that tinnitus more frequently and more intensely seems to develop in patients with lower serum values of Vit D should persuade patients and clinicians to routinely assess Vit D status in patients with acute and especially chronic tinnitus. The identification of a low serum Vit D concentration will enable to correct the deficiency, not only ameliorating tinnitus but also lowering the risk of developing a large number of health disorders that frequently accompany Vit D deficiency (i.e., osteoporosis, cardiovascular and autoimmune diseases, infections, cancer, metabolic syndrome and diabetes among others) [47,48]. As concerns the specific management of tinnitus, the identification of the underlying cause remains elusive in a large number of patients, so that the treatment remains mostly symptomatic (i.e., encompassing psychotherapy, psychoactive drugs administration, physical therapy, use of individualized sound stimulation or masking devices, cognitive behavioral or tinnitus retraining therapy) and not completely resolutive in the vast majority of cases even when a possible cause can be identified [49]. Although large randomized clinical trials on Vit D supplementation in patients with tinnitus are still unavailable to the best of our knowledge, the recent evidence emerged from the study of Abdelmawgoud Elsayed [30], that Vit D supplementation was accompanied by substantial reduction of mental and physical impairment due to idiopathic tinnitus, leads the way to explore the possibility of administering Vit D to all patients with tinnitus and with concomitantly low serum levels of this important vitamin. It is hence advisable that future searches, including more studies and from a more widespread field of research, will be done that in the future, thus allowing to provide more solid evidence on this matter.

## 5. Conclusions

Tinnitus is a frequently disabling condition characterized by appearance of ringing or other noises in the ears, that are not typically generated by an external sound. Vit D is instead an essential nutrient, which plays a major role in a variety of bodily functions, some of these directly connected with the hearing function. It is hence not surprising that the results emerged from out critical literature review and meta-analysis indicate that lower serum Vit D levels could be associated with tinnitus. These findings pave the way to plan future randomized prospective trials aimed at exploring as to whether Vit D supplementation may aid in preventing and/or reducing tinnitus-related impairment, irrespective or not of the potential benefits in preventing underlying medical conditions associated with an enhanced risk of developing tinnitus.

## Figures and Tables

**Figure 1 diagnostics-13-01037-f001:**
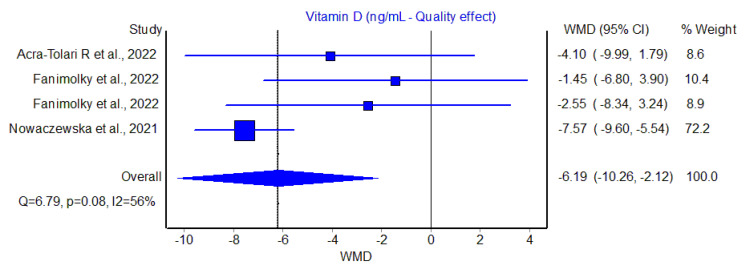
Weighted mean difference (WMD) and 95% confidence interval (95% CI) of serum vitamin D (Vit D) values in patients with or without tinnitus [16,17,18].

**Figure 2 diagnostics-13-01037-f002:**
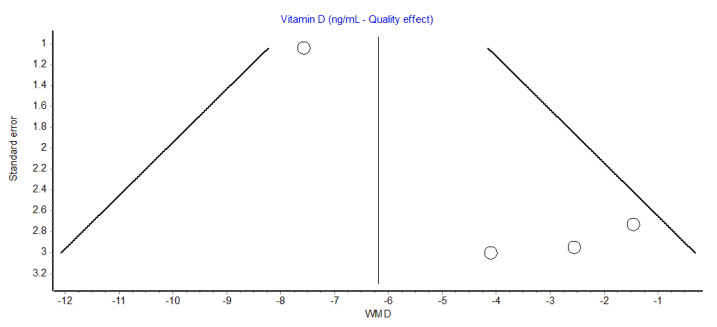
Funnel plot of observational studies reporting serum vitamin D concentration in patients with or without tinnitus. WMD, weighted mean difference.

**Table 1 diagnostics-13-01037-t001:** Main characteristics of studies reporting serum vitamin D concentration in patients with or without tinnitus.

Authors	Setting	Study Design	Sample Size	Age and Sex	Vitamin D Assessment
Acra-Tolari R et al., 2022 [16]	Dominican Republic	Observational, case-control	Postmenopausal women with tinnitus (5) or with history of hearing difficulty without tinnitus (*n* = 39)	Entire cohort: median age, 74 (IQR, 12) years and 100% women	Not described
Fanimolky et al., first cohort 2022 [17]	Iran	Observational, case-control	Ear cholesteatoma patients with (*n* = 43) or without (*n* = 19) tinnitus	Entire cohort: mean age, 32 ± 1 years and 34% females	ELISA
Fanimolky et al., second cohort 2022 [17]	Iran	Observational, case-control	Chronic otitis media patients with (*n* = 19) or without (*n* = 43) tinnitus	Entire cohort: mean age, 34 ± 1 years and 34% females	ELISA
Nowaczewska et al., 2021 [18]	Poland	Observational, case-control	Patients with tinnitus (*n* = 201) and matched healthy controls (*n* = 99)	Tinnitus cohort: mean age, 49.9 ± 13.2 years and 54% females; control cohort: mean age, 48.3 ± 17.5 years and 52% females	Not described

ELISA, Enzyme-linked immunosorbent assay; IQR, interquartile range.

**Table 2 diagnostics-13-01037-t002:** Mechanisms by which low serum vitamin D levels may trigger or aggravate tinnitus.

❖ Increased risk of tympanosclerosis ❖ Higher likelihood of developing acute or chronic middle ear pathologies ▪ Acute or chronic otitis media ▪ Otosclerosis ▪ Cochlear demineralization ❖ Enhanced odds of hypertension ❖ Increased risk of advancing age deafness and presbycusis ❖ Major perception of pre-existing tinnitus ▪ Demineralization of petrous temporal bone ▪ Stress, anxiety and depression

## Data Availability

Not applicable.

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
