# Peer review of "Serum Vitamin D Concentration Is Lower in Patients with Tinnitus: A Meta-Analysis of Observational Studies"

_diagnostics, 2023, doi:10.3390/diagnostics13061037_

Round 1
Reviewer 1 Report (Previous Reviewer 1)
I agree with both your points and corrections.
Author Response
I agree with both your points and corrections.
- We are thankful to the referee. No additional comments to be addressed.
Reviewer 2 Report (Previous Reviewer 2)
The link between Vitamin D deficiency and hearing problems and tinnitus is relevant, and bringing this to the attention of the scientific community is important, but I think this article must be ammeliorated first.
The choice of mixing Conductive hearing losses and Sensorineural hearing losses does not seem convincing, or the relationship and pathophysiology between Vitamin D deficiency and different types of hearing losses should be more thoroughly explained (not just pointing to Rickets). I specially look forward to hear where the relationship to Colesteatoma stands.
Overall, I think a major revision is advised.
Author Response
The link between Vitamin D deficiency and hearing problems and tinnitus is relevant, and bringing this to the attention of the scientific community is important, but I think this article must be ammeliorated first.
- We are thankful to the referee for the globally favourable comments on our manuscript. We’ll do our best to improve it according to the referee’s suggestions.
The choice of mixing Conductive hearing losses and Sensorineural hearing losses does not seem convincing, or the relationship and pathophysiology between Vitamin D deficiency and different types of hearing losses should be more thoroughly explained (not just pointing to Rickets).
- This is a good point. The entire part of conductive hearing losses and sensorineural hearing losses has been amended to copy with this comment (highlighted in the text).
I specially look forward to hear where the relationship to Colesteatoma stands.
- This is a good point. The part on Colesteatoma has been amended to copy with this comment (highlighted in the text).
Round 2
Reviewer 2 Report (Previous Reviewer 2)
Dear Authors
I am pleased to see that the recommended alterations to the manuscript were answered. The article was already quite interesting, and the alterations made the explanation more thorough.
I could accept it for publishing.
This manuscript is a resubmission of an earlier submission. The following is a list of the peer review reports and author responses from that submission.
Round 1
Reviewer 1 Report
This is a mata-analysis introducing the role of vitamin D in the development of tinnitus based on four groups of qualified study. Both methods and discussions were in accordance with the submission guidelines of the journal. The conclusions were the same as most physicians know from the points of literature and clinics. This will help doctors keep in mind of the story behind the clinical phenomenon of tinnitus.
It's open to discuss the novelty of the manuscript. Afterall physicians need the latest news as well as updated knowledge to arm themselves in terms of research and clinical tools. In this aspect comparisons between results from this study and others from literature found scarce new hints.
It's hoped that in the future a further search including more studies and from a more widespread field of research will be done.
Reviewer 2 Report
Dear editor/ authors
The main question addressed by the research is the relevance of the relationship of vitamin d and tinnitus, but in my opinion that is not what is interesting about this article. The main contribution of this article is the meta analysis. And it is of utmost relevance the analysis of the type of research used. The article explains the physiopathology of tinnitus and previous research, but I think that is not what is most interesting here. Therefore the analysis and explaining of the graphic is important and the results of the meta analysis itself, not the physiopathology. Explaining the articles revised, their methodology and relevance to the meta analysis is also important and since there are only 4 articles it is quite easy. The text is well written but I want to see the graphic and results better explained. The conclusion will only be consistent to the results if the results are better explained. The main question should be: is vitamin d relevant to the problem of tinnitus regarding the meta analysis of relevant current research ? Not regarding the physiopathology of the disease.
overall the article is relevant and well written, but the most important part of this kind of review is the results based on the forest plot graphic, which I felt could be better explained. This is a must do in my opinion